# New Carboxytriazolyl Amphiphilic Derivatives of Calix[4]arenes: Aggregation and Use in CuAAC Catalysis

**DOI:** 10.3390/ijms242316663

**Published:** 2023-11-23

**Authors:** Diana Mironova, Ilshat Bogdanov, Aliya Akhatova, Elza Sultanova, Ramilya Garipova, Artur Khannanov, Vladimir Burilov, Svetlana Solovieva, Igor Antipin

**Affiliations:** 1Alexander Butlerov Institute of Chemistry, Kazan Federal University, 18 Kremlevskaya Str., 420008 Kazan, Russia; 2Arbuzov Institute of Organic and Physical Chemistry, FRC Kazan Scientific Center of RAS, 8 Arbuzov Str., 420088 Kazan, Russia

**Keywords:** calixarene, aggregation, CuAAC, click reaction, 1,4-triazole

## Abstract

This work focuses on the synthesis of a new series of amphiphilic derivatives of calix[4]arenes for the copper(I)-catalyzed azide-alkyne cycloaddition (CuAAC) reaction. The aggregation properties of synthesized calix[4]arenes were studied using various techniques (fluorescence spectroscopy, nanoparticle tracking analysis, and dynamic light scattering). Increasing the length of the alkyl substituent led to stronger hydrophobic interactions, which increased polydispersity in solution. The zwitterionic nature of the synthesized calix[4]arenes was established using different types of dyes (Eosin Y for anionic structures and Rhodamine 6G for cationic structures). The synthesized calix[4]arenes were used as organic stabilizers for CuI. The catalytic efficiency of CuI-calix[4]arene was compared with that of the phase transfer catalyst tetrabutylammonium bromide (TBAB) and the surfactant sodium dodecyl sulfate (SDS). For all calixarenes, the selectivity in the CuAAC reaction was higher than that observed when TBAB and SDS were estimated.

## 1. Introduction

In the modern world, catalysis plays a crucial role since most chemical processes are catalytic. Strict environmental and economic requirements dictate the importance of creating new, technologically useful, efficient, selective catalysts for the oil, industrial, and medical fields. The development of new efficient catalytic systems is the main task in the field of click chemistry [1]. Click reactions are a universal approach for the design of complex molecular systems. The copper(I)-catalyzed azide-alkyne cycloaddition reaction (CuAAC) is currently under active research. The advantage of the CuAAC reaction is the availability of reaction substrates, the mild reaction conditions involved, and the wide variety of applications for the obtained molecules, namely, pharmaceuticals [2], new materials [3,4], and polymers [5,6]. The disadvantages of this reaction are the need for a large catalyst load as well as the need to stabilize copper in the (I) oxidation state [7,8]. The main approach to improving the CuAAC reaction is to use different combinations of copper with ligands, that is nanoreactors [7,8,9], complexes [10], and polymer substrates [11,12]. Despite the wide variety of applications of CuAAC, works in which a low catalyst loading is combined with high activity are rare. One of the principles of click-chemistry is the complete avoidance of solvents or the use of a harmless water solvent [13,14,15]. The most common ligand used in CuAAC is tris(benzyltriazolemethyl)amine (TBTA) [16,17]. Currently, various derivatives of TBTA (tris((1-hydroxy-propyl-1H-1,2,3-triazol-4-yl)methyl)amine (THPTA), 2-[4-((bis[(1-tert-butyl-1H-1,2,3-triazol-4-yl)methyl]amino)methyl)-1H-1,2,3-triazol-1-yl]acetic acid (BTTAA), 2-[4-((bis[(1-tert-butyl-1H-1,2,3-triazol-4-yl)methyl]amino)-methyl)-1H-1,2,3-triazol-1-yl]ethyl hydrogen sulfate (BTTES), tris[(1-(2-ethoxy-2-oxoethyl)-1H-1,2,3-triazol-4-yl)methyl]amine (TEOTA), 3-[4-((bis[(1-tert-butyl-1H-1,2,3-triazol-4-yl)methyl]amino)methyl)-1H-1,2,3-triazol-1-yl]propanol (BTTP), and 3-[4-((bis[(1-tert-butyl-1H-1,2,3-triazol-4-yl)methyl]amino)methyl)-1H-1,2,3-triazol-1-yl]propyl hydrogen sulfate (BTTPS), which have better water solubility and less toxicity and can also effectively stabilize Cu(I), are widely used [18,19,20,21].

Calixarenes have great potential in coordinating Cu in the CuAAC reaction. They are macrocyclic compounds consisting of phenolic rings that are easily functionalized at the upper or lower rims. The synthetic availability of calixarenes allows for the creation of an unlimited number of structures with given fragments, allowing researchers to vary their properties [22,23,24]. Thus, N-heterocyclic carbene complexes of Cu(I) on a calixarene platform demonstrated good activity in the CuAAC reaction [25,26]. The use of calixarene amphiphiles makes it possible to significantly improve the catalytic activity of metal salts both due to metal stabilization and the micellar effect, as previously shown in the example of C-C cross-coupling reactions catalysis [27,28,29,30].

In the present work, we report the synthesis of new amphiphilic zwitterionic derivatives of calix[4]arenes containing two carboxytriazole moieties. The resulting calix[4]arene derivatives were tested as a micellar medium in the CuAAC reaction. The aggregation characteristics were studied via fluorescence, UV-visible, dynamic light scattering (DLS), and nanoparticle tracking analysis (NTA) methods. The obtained calixarenes were used as micellar catalysts in a one-pot CuAAC reaction of benzyl/alkyl halides with phenylacetylene in the presence of NaN_3_ and CuI. A comparative analysis of the selectivity of the obtained calixarenes with conventional surfactants in the CuAAC reaction was carried out.

## 2. Results and Discussion

### 2.1. Synthesis

Calix[4]arene derivatives **5a**,**b** and **6a**,**b** containing ammonium/imidazolium moieties with propyl azide groups on the upper rim and butyl/octyl substituents on the lower rim were synthesized. The methodology for the synthesis of the target products was previously described in [31]. For this purpose, compound **2** was synthesized from the starting *p-tert*-butylcalix[4]arene **1** in the presence of phenol and aluminum (III) chloride, which were further introduced into the Williamson reaction with *n*-butyl/*n*-octyl bromide to yield the di-O-alkyl derivatives **3a**–**b**. The macrocycles **3a**–**b** were then modified along the upper rim with chloromethyl groups to produce bis-chloromethylated calix[4]arene derivatives **4a**–**b**. Further, the halogen-containing macrocycles **4a**–**b** were reacted with N-(3-azidopropyl)-N,N-dimethylamine as well as 1-(3-azidopropyl)-1H-imidazole in acetonitrile at 80 °C. The spectral characteristics of the obtained ammonium and imidazolium salts of calix[4]arene **5a**–**b** and **6a**–**b** agree with the previously published data [31]. In the final stage, the azide-containing derivatives **5a**–**b** and **6a**–**b** were used in an AAC reaction with 5 eq. of acetylenedicarboxylic acid in toluene at 60 °C. Since the structure of alkyne has acceptor carboxyl groups, a cycloaddition reaction occurs without the use of copper catalysis. Thus, bis-4,5-dicarboxylate-triazole ammonium/imidazolium macrocycles **7a**–**b** and **8a**–**b** were obtained in 58–78% yields (Figure 1). The structures of the new compounds were characterized using various physical methods, including ^1^H- and ^13^C-NMR, IR spectroscopy, and ESI high-resolution mass spectrometry.

Due to the increased acidity, the protons of carboxyl groups are subject to deuteroexchange with solvent molecules, so these protons do not appear in the ^1^H NMR spectrum. Moreover, due to the amphiphilicity of macrocycles, broadening of all the signals can be observed in the ^1^H NMR spectrum. The ^1^H NMR spectrum of compound **7a** (Appendix A) shows signals from the protons of aromatic rings as a singlet, doublet, and triplet at δ 7.21 ppm, 7.01 ppm (6.6 Hz coupling constant), and 6.81 ppm (6.4 Hz coupling constant), respectively. The broadened singlet at δ 4.82 ppm corresponds to the signals of protons of methylene fragments bound by triazole nuclei. The protons of methyl groups beside nitrogen atoms appear as a singlet at δ 2.84 ppm. In the IR spectra of compounds **7a**–**b** and **8a**–**b**, the absence of an intense absorption band at 2100 cm^−1^ corresponding to the valence asymmetric vibrations of the azide group is most informative (Appendix A). After click reactions, the ^13^C-NMR spectra of compounds **7a**–**b** and **8a**–**b** show four signals in the 160–167 ppm region generated by magnetically non-equivalent carboxyl carbon atoms and carbon atoms of the triazole ring (Figure 1). Typically, signals from carbons 4 and 5 of the triazole ring have a chemical shift in the 124 ppm and 143 ppm regions, respectively. However, in the case of compounds **7a**–**b** and **8a**–**b**, the triazole carbon atoms are associated with electron-withdrawing carboxyl groups, as a result of which the signals of these atoms are shifted to the low-field region. In the ESI high-resolution mass spectra of compounds **7a**–**b** and **8b,** the signals of doubly charged molecular ions have low intensity (*m*/*z* 579.3177 [M-2Cl]^2+^ was found for compound **7b**, calculated for C_64_H_86_N_8_O_122_^+^ 579.3128), and in the case of compound **8a**, no molecular ion signal was observed. The most intense signals in the mass spectra were from various chinonmethide structures formed as a result of the cleavage of ammonium/imidazolium fragments (Appendix A). It was shown earlier for imidazolium derivatives of calix[4]arene that when using a mild ionization method such as electrospray, signals from various quinomethide structures can also be observed in high-resolution mass spectra in addition to the peaks of molecular doubly charged ions [32].

### 2.2. Aggregation Properties

The new calixarenes **7a**,**b** and **8a**,**b** are soluble in a TRIS buffer (200 mM; pH 7.0). The presence of alkyl substituents on the lower rim and polar fragments on the upper rim indicate the amphiphilic properties of the studied calixarenes. Therefore, critical aggregation concentrations (CACs) were evaluated in the presence of a fluorescent probe, pyrene [33]. The measured CAC values are listed in Table 1. For the two types of calixarenes (bearing imidazole or an ammonium moiety), the same tendency was observed: as the length of the alkyl substituent increases, the CAC values decrease. This effect was previously observed for other isostructural imidazolium calix[4]arenes with octyl and butyl alkyl substituents at the lower rim and alkynyl and azide moieties at the upper rim [32,34]. The same results for amphiphilic calix[4]arene derivatives with a 4,5-dicarboxytriazolyl fragment directly related to the macrocyclic platform was observed [29]. This may be due to the fact that with an increasing number of carbon atoms in the alkyl substituents, the contribution of hydrophobic interactions increases, which leads to the earlier formation of aggregates.

The synthesized calixarenes are close to zwitterionic active substances in terms of their structure since they consist of both negative (carboxylate fragments) and positive (imidazole or ammonium) fragments separated by a short methylene -(CH_2_)_3_- segment. In aqueous solutions, zwitterionic micelles are known to preferentially solubilize anionic structures [35,36,37]. According to the UV-visible spectroscopy results, the obtained zwitterionic calixarenes interact more effectively with the anionic dye Eosin Y (EY) compared to the cationic dye Rhodamine 6G (Rh6G) (Appendix A), confirming the greater contribution of the positive charge in the compounds’ aggregates. Thus, in the presence of small amounts of **7a**–**b** and **8a**–**b**, the absorption maximum of Eosin Y shifts to a longer wavelength (Δ 3–5 nm). The maximum shift to the long-wavelength region (Δ 12.5–15.5 nm) was observed after CAC (Appendix A). At the same time, for the cationic Rhodamine 6G, a bathochromic shift in the absorption band maximum was not observed in the whole range of investigated concentrations (Appendix A).

The standard method for estimating the size of aggregates formed by macrocycles is the dynamic light scattering (DLS) method. However, in the case of **7a**–**b** and **8a**–**b**, we observed high values of polydispersity and large disorder (Appendix A). Therefore, nanoparticle tracking analysis (NTA) was used to adequately evaluate the aggregation properties of the macrocycles.

NTA showed that in the initial solution, a set of particles with average polydispersity had been formed (Figure 2A); within 24 h, the destruction of primary aggregates occurred, both in the direction of decreasing particle size and in the direction of enlargement (Figure 2B). Within a week, the solution was completely stabilized, and in it, there were two bases, including particles with sizes of 76 and 108 nm, and their associate with a diameter of 170 nm; thus, the solution became monodisperse (Figure 2C).

In contrast to **8b**, the self-organization and stabilization of **7b** proceed through a phase consisting of the formation of a large number of smaller particles, predominantly particles with hydrodynamic diameters of 40 and 60 nm (Appendix A). On the seventh day, the system was stabilized but with more polydispersion in size, with two main fractions of nanoparticles 94 and 140 nm present in the solution (Appendix A), but unlike compound **8b**, **7b** does not form joint associates. But dimers of the main fraction with hydrodynamic diameters of 206 (dimer 94) and 303 (trimer 94 or dimer 140) were present in the solution (Table 2). In the case of compound **8a**, the solution showed the highest polydispersity when dissolved, with the inability to adequately detect the resulting associates (Appendix A); the solution contained very large aggregates. After 24 h, the solution was stabilized for NTA detection but showed extremely high polydispersity (Appendix A). For 7 days, no stabilization of the particles in the system was observed (Appendix A). The main fractions remained the same, and there was a slight increase in the concentration of fractions with hydrodynamic diameters of 114, 163, and 189 nm (Table 2), probably due to the aggregation of smaller particles. For **7a**, reliable results obtained via the NTA method could only be obtained on day 7; before that, micrometer-range agglomerates were present in the solution, and these do not allow for a reliable determination of particle size (Appendix A). On the seventh day, the solution was stabilized (Appendix A) to obtain reliable results, but the polydispersity in it was higher than that in a similar solution of compound **8a**.

### 2.3. Catalysis

Calixarenes **7a**–**b** and **8a**–**b** were used as organic substrates for CuI. We began this study with a model CuAAC reaction of phenylacetylene, benzyl chloride, and sodium azide. The catalytic effect of calixarene + Cu(I) systems was studied at two concentrations of calixarene: above CAC (0.1 mM) and below CAC (0.001 mM). In both cases, the load of CuI was 0.5 mole %. The reaction was incubated at 40 °C for 6 h in water. The obtained results are summarized in Table 3. Conversion in the reaction was determined according to halogen derivative intake using gas chromatography–mass spectrometry (GCMS) with the absolute calibration method; selectivity was determined according to the ratio of the target product to the 1,5-triazole products. Compared to free CuI, all the in situ formed calixarene + Cu(I) complexes led to higher yields of 1,4-triazole. A 1.5-fold increase in efficiency was also observed when switching from calixarenes with butyl alkyl moieties (**7a** and **8a**) to more lipophilic-octyl moieties (**7b** and **8b**). This result was a consequence of the micellar effect of the systems used: calixarenes not only form metal complexes in situ but also function as nanoreactors that concentrate reagents. This allows reactions with water-insoluble substrates to be carried out in water. When comparing the efficiency of CuI in the presence of the well-known interfacial catalyst tetrabutylammonium bromide (TBAB) [38,39] and the classical surfactant sodium dodecyl sulfate (SDS) [40,41], it can be seen that **7b** is not inferior in efficiency. Its excellent catalytic abilities were attributed to the synergetic effect of charged fragments and cavities of the calixarene skeleton [42,43], where functionalized macrocycles unite the properties of the phase-transfer catalyst and micellar nanoreactor. The amphiphilic calixarenes are capable of both solubilizing and transferring hydrophobic substrates and the active center (copper iodide) from the aqueous medium into the cavity of the calixarene platform as well as into the hydrophobic part of the formed associates.

Nevertheless, the influence of the structure of the hydrophilic part of the calixarenes remains unclear. For this reason and to increase the product yield, the optimal parameters for two types of catalytic systems, **7b** + CuI and **8b** + CuI, were screened, and the results are presented in Table 4. Varying the amount of CuI revealed that the best efficiency could be obtained at a 0.5 mole % loading (Table 4), which is probably related to the optimal stabilization of Cu ions on calixarenes. Increasing the temperature led to an increase in the yield of the by-product 1,5-substituted triazole in all systems. Consequently, the optimal reaction conditions are 0.5 mole % CuI applied to the reactants and a temperature of 40 °C.

Calixarene **7b** was selected as an organic substrate in the in situ complexes to demonstrate the versatility of the optimized CuAAC reaction conditions in an aqueous medium for halogen derivatives of different structures. The results are summarized in Table 5.

The calixarene (**7b**)-based system showed satisfactory results in catalyzing “one-pot” click reactions for halide derivatives. Even better results were obtained for the alkyl-containing bromides, which have low reactivity [44], which is due to their potential for stabilization and fixed orientation due to the hydrophobic effect with the macrocycle.

## 3. Materials and Methods

Solvents were purified according to standard procedures. Syntheses were performed using commercially available reagents obtained from Sigma-Aldrich, Alfa-Aesar, and Maclin catalogs.

25,26,27,28-tetrahydroxycalix[4]arene **2** [45], 25,27-dihydroxy-26,28-dibutoxycalix[4]arene **3a**, 25,27-dihydroxy-26,28-dioctyloxycalix[4]arene **3b**, 11,23-dichloromethyl-25,27-dihydroxy-26,28-di-butoxycalix[4]arene **4a**, 11,23-dichloromethyl-25,27-dihydroxy-26,28-dioctyloxycalix[4]arene **4b**, 11,23-bis[(3-azidopropyl)dimethylammonium)me-thyl]-25,27-dihydroxy-26,28-dibutoxycalix[4]arene dichloride **5a**, 11,23-bis[(3-azidopropyl)dimethylammonium)me-thyl]-25,27-dihydroxy-26,28-dioctyloxycalix[4]arene dichloride **5b**, 11,23-bis[3(1-(3-azidopropyl))-1H-imidazolium]-25,27-dihydroxy-26,28-dibutoxycalix[4]arene dichloride **6a**, 11,23-bis[3(1-(3-azidopropyl))-1H-imidazolium]-25,27-dihydroxy-26,28-dioctyloxycalix[4]arene dichloride **6b** [31], 3-azido-N,N-dimethylpropane-1-amine [46], and 1-(3-azidopropyl)-1H-imidazole [47] were obtained in accordance with methods from the literature.

The purity of substances was controlled using TLC conducted on Merck UV 254 plates and observed using UV light produced by a VL-6.LC lamp. NMR experiments were performed at room temperature using Bruker’s Avance 400 Nanobay spectrometer. IR spectra of the obtained compounds were recorded using a Bruker Vector-22 FT-IR spectrometer in the wave number range of 400–4000 cm^−1^ in KBr pellets. HRMS-ESI mass spectra were obtained using an Agilent iFunnel 6550 LC/Q-TOF mass spectrometer. Gas chromatography–mass spectrometry was performed using a GCMS-QP2010 Ultra gas chromatography mass spectrometer (Shimadzu, Kyoto, Japan) equipped with an HP-5MS column (for which the internal diameter was 0.32 mm, and the length was 30 m). The parameters were as follows: Helium 99.995% purity was using as the carrier gas, the temperature of the injector was 250 °C, the flow rate through the column was 2 mL/min, and the thermostat temperature program was a gradient temperature increase from 70 to 250 °C with a step of 10 °C/min. The range of the scanned masses was *m*/*z* 35–400. The internal standard method using dodecane was used for quantitative analysis.

### 3.1. General Procedure for the Preparation of Calixarene Derivatives ***7a***–***b*** and ***8a***–***b***

To 0.11 mmol of compound **5a**/**5b**/**6a**/**6b**, 0.56 mmol of acetylenedicarboxylic acid and 4 mL of toluene were added. The reaction mixture was stirred at 60 °C. After the set time elapsed, the solvent was removed under reduced pressure. After evaporation, unreacted acetylenedicarboxylic acid remained in the reaction product. To remove it, the target compound was washed with 20 mL of diethyl ether. In this case, the reaction product was precipitated, filtered, and dried in vacuo.


**11,23-bis[(3-(4,5-dicarboxy-1,2,3-triazol-1-yl)propyl)dimethylammonium)methyl]-25,27-dihydroxy-26,28-dibutoxycalix[4]arene dichloride (7a)**


Reaction time 60 h. Yield 0.075 g (61%). mp = 168 °C (decomp). NMR ^1^H (400 MHz, DMSO-d_6_/D_2_O = 3/1, 25 °C) δH, ppm.: 1.03 (t, J = 6.9 Hz, 6H, CH_3_), 1.67–1.83 (m, 4H, CH_2_), 1.92 (br s, 4H, CH_2_), 2.36 (br s, 4H, CH_2(am)_), 2.84 (s, 12H, CH_3(am)_), 3.10 (br s, 4H, CH_2_), 3.48 (br d, 4H, Ar-CH_2_-Ar), 3.95 (br s, 4H, O-CH_2_), 4.07–4.13 (m, 4H, CH_2(am)_), 4.23 (br s, 4H, Ar-CH_2_-Ar), 4.82 (s, 4H, CH_2_-Im), 6.80 (t, J = 6.4 Hz, 2H, HAr), 7.03 (d, J = 6.6 Hz, 4H, HAr), 7.20 (s, 4H, HAr). NMR ^13^C-[^1^H] (100.9 MHz, DMSO-d6, 25 °C) δC, ppm.: 165.38, 162.60, 160.95, 159.21, 154.45, 151.80, 140.81, 133.43, 132.83, 129.11, 128.26, 125.84, 125.09, 118.24, 76.48, 49.22, 47.01, 42.16, 39.52, 31.82, 30.24, 18.88, 13.98. IR (KBr) ν_max_, cm^−1^: ν 1461.84 (C=C), ν 1719.94 (-C(O)), ν_as_ 2933.50 (CH_2_), ν_s_ 2872.76 (CH_2_), ν 3275.22 (OH). HRMS-ESI: found to be *m*/*z* 803.4007 [M-2Cl-C_10_H_16_N_4_O_4_]^+^; calculated for C_47_H_55_N_4_O_8_^+^ 803.4014; found to be *m*/*z* 523.2550 [M-2Cl]^2+^; calculated for C_56_H_70_N_8_O_12_^2+^ 523.2551; found to be *m*/*z* 583.2823 [M-2Cl-C_18_H_30_N_8_O_8_-2Cl+Na]^+^; calculated for C_38_H_40_NaO_4_^+^ 583.2819; found to be *m*/*z* 561.3002 [M-2Cl-C_18_H_29_N_8_O_8_-2Cl]^+^; calculated for C_38_H_41_O_4_^+^ 561.2999.


**1,23-bis[(3-(4,5-dicarboxy-1,2,3-triazol-1-yl)propyl)dimethylammonium)methyl]-25,27-dihydroxy-26,28-dioctyloxycalix[4] arene dichloride (7b)**


Reaction time 57 h. Yield 0.1 g (74%). mp = 166 °C (decomp). NMR ^1^H (400 MHz, DMSO-d_6_/D_2_O = 3/1, 25 °C) δH, ppm: 0.79 (br s, 6H, CH_3_), 1.15–1.49 (m, 16H, CH_2_), 1.71 (br s, 4H, CH_2_), 1.94 (br s, 4H, CH_2_), 2.38 (br s, 4H, CH_2(am)_), 2.84 (br s, 12H, CH_3(am)_), 3.11 (br s, 4H, CH_2_), 3.48 (br d, 4H, Ar-CH_2_-Ar), 3.94 (br s, 4H, O-CH_2_), 4.24 (br s, 4H, Ar-CH_2_-Ar), 4.82 (s, 4H, CH_2_-Im), 6.81 (br s, 2H, HAr), 7.01 (br d, 4H, HAr), and 7.21 (s, 4H, HAr). NMR ^13^C-[^1^H] (100.9 MHz, DMSO-d_6_, 25 °C) δC ppm: 165.07, 162.76, 161.01, 159.23, 154.49, 151.81, 133.45, 133.27, 129.09, 128.30, 126.65, 118.25, 76.71, 75.06, 49.22, 46.97, 42.17, 31.47, 30.25, 29.63, 29.00, 28.88, 25.50, 23.30 22.20, and 14.00. IR (KBr) ν_max_, cm^−1^: ν 1461.52 (C=C), ν 1719.79 (-C(O)), ν_s_ 2856.13 (CH_2_), and ν_as_ 2928.09 (CH_2_), ν 3277.61 (OH). HRMS-ESI: found to be *m*/*z* 579.3184 [M-2Cl]^2+^; calculated for C_64_H_86_N_8_O_12_^2+^ 579.3177; found to be *m*/*z* 673.4255 [M-2Cl-C_18_H_29_N_8_O_8_-2Cl]^+^; calculated for C_46_H_57_O_4_^+^ 673.4251; found to be *m*/*z* 695.4067 [M-2Cl-C_20_H_30_N_8_O_8_+Na]^+^; calculated for C_46_H_56_NaO_4_^+^ 695.4071; found to be *m*/*z* 915.5266 [M-2Cl-C_9_H_15_N_4_O_4_]^+^; calculated for C_46_H_56_NaO_4_^+^ 915.5266.


**11,23-bis[(1-(3-(4,5-dicarboxy-1,2,3-triazol-1-yl)propyl))-1H-imidazolium)methyl]-25,27-dihydroxy-26, 28-dibutoxycalix[4]arene dichloride (8a)**


Reaction time 60 h. Yield 0.075 g (58%). mp = 162 °C (decomp). NMR ^1^H (400 MHz, DMSO-d_6_/D_2_O = 3/1, 25 °C) δH, ppm: 0.98 (t, J = 6.8 Hz, 6H, CH_3_), 1.71 (br s, 4H, CH_2_), 1.88 (br s, 4H, CH_2_), 2.29–2.41 (br m, 4H, CH_2_), 3.40 (d, J = 11.9 Hz, 4H, Ar-CH_2_-Ar), 3.88 (br s, 4H, O-CH_2_), 4.07 (d, J = 12.7 Hz, 4H, Ar-CH_2_-Ar), 4.44 (t, J = 6.0 Hz, 4H, CH_2_), 4.75 (t, J = 6.2 Hz, 4H, CH_2_), 5.13 (s, 4H, CH_2_-Im), 6.60–6.73 (br m, 2H, HAr), 6.95 (br d, 4H, HAr), 7.20 (s, 4H, HAr), and 7.50–7.65 (br m, 4H, HIm). NMR ^13^C-[^1^H] (100.9 MHz, DMSO-d_6_/D_2_O = 3/1, 25 °C) δC ppm: 166.59, 164.21, 162.47, 160.41, 153.89, 152.60, 138.44, 136.37, 134.21, 129.86, 129.20, 126.48, 125.71, 123.36, 123.09, 77.24, 52.77, 47.73, 47.20, 32.55, 31.01, 30.52, 19.67, and 14.66. IR (KBr) ν_max_, cm^−1^: ν 1460.10 (C=C), ν 1710.88 (-C(O)), ν_as_ 2931.24 (CH_2_), ν_s_ 2871.69 (CH_2_), ν 3307.63 (OH). HRMS-ESI: found to be *m*/*z* 894.4181 [M-2Cl-C_7_H_8_N_3_O_4_]^+^; calculated for C_51_H_56_N_7_O_8_^+^ 894.4185; found to be *m*/*z* 629.3379 [M-2Cl-C_17_H_19_N_8_O_8_]^+^; calculated for C_41_H_45_N_2_O_4_^+^ 629.3374.


**11,23-bis[(1-(3-(4,5-dicarboxy-1,2,3-triazol-1-yl)propyl))-1H-imidazolium)methyl]-25,27-dihydroxy-26,28-dioctyloxycalix[4]arene dichloride (8b)**


Reaction time 54 h. Yield 0.11 g (78%). mp = 165 °C (decomp). NMR ^1^H (400 MHz, DMSO-d_6_/D_2_O = 3/1, 25 °C) δH ppm: 0.76 (br s, 6H, CH_3_), 1.13–1.42 (br m, 16 H, CH_2_), 1.66 (br s, 4 H, CH_2_), 1.90 (br s, 4H, CH_2_), 2.33 (br s, 4H, CH_2_), 3.41 (br d, 4H, Ar–CH_2_–Ar), 3.87 (br s, 4H, O-CH_2_), 4.16 (br d, 4H, Ar–CH_2_–Ar +CH_2_), 4.73 (br s, 4H, CH_2_), 5.10 (br s, 4H, Ar-CH_2_-Im), 6.66 (br s, 2H, HAr), 6.95 (br s, 4H, HAr), 7.19 (br s, 4H, HAr), and 748–7.60 (br m, 4H, HIm). NMR ^13^C-[^1^H] (100.9 MHz, DMSO-d_6_, 25 °C) δC ppm.: 165.26, 162.53, 161.10, 159.19, 153.13, 151.79, 140.81, 136.09, 133.49, 132.67, 129.01, 128.31, 125.58, 125.02, 122.75, 122.52, 76.76, 51.79, 46.69, 46.43, 31.45, 30.30, 30.03, 29.61, 28.98, 28.87, 25.45, and 22.19. IR (KBr) ν_max_, cm^–1^: ν 1459.90 (C=C), ν 1485.42 (C-C), ν 1717.59 (-C(O)), ν_s_ 2856.06 (-CH_2_-), ν_as_ 2928.19 (-CH_2_-), and ν 3294.53 (OH). HRMS-ESI: found to be *m*/*z* 602.2967 [M-2Cl]^2+^; calculated for C_66_H_80_N_10_O_12_^2+^ 602.2973.

### 3.2. Sample Preparation

All solutions were prepared using MilliQ water (resistivity 0.055 microSiemens). The stock solutions of calixarenes and dyes were prepared in 200 mM of TRIS-HCl (tris(hydroxymethyl)aminomethane) buffer. The pH of the solutions was maintained constant at 7.4.

### 3.3. Steady-State Fluorescence Study

CAC values were measured using pyrene fluorescent probe and calculated from the dependence of the intensity ratio of the first (373 nm) and third (384 nm) bands in the emission spectrum of pyrene vs. calixarene concentration. Fluorescence experiments with pyrene were performed in 10.0 mm quartz cuvettes and recorded in the range of 340 to 430 nm at an excitation wavelength of 335 nm with a 2.5 nm slit. All studies were conducted in buffered aqueous solution (TRIS buffer, pH 7.4) at 25 °C.

### 3.4. UV-Visible Absorption Spectroscopic Study

The spectra between the calixarenes and dyes were obtained. The spectra were recorded between 400 and 600 nm, using a quartz cuvette of path length 1 cm.

### 3.5. Dynamic Light Scattering Study

DLS experiments were carried out using Zetasizer Nano ZS instrument (Malvern Panalytical, Worcestershire, UK) with 4 mW 633 nm He–Ne laser light source and a light-scattering angle of 173. The data were processed using DTS software (Dispersion Technology Software 5.00). The solutions were filtered through 800 nm filter before the measurements to remove dust. The experiments were carried out in disposable plastic cells, DTS 0012, at 25 °C, with at least three experiments for each system. Statistical data treatment was performed using *t*-Student coefficient, and the particle size determination error was <2%.

### 3.6. Nanoparticle Tracking Analysis

NTA analysis was performed using a NanoSight LM-10 instrument (Malvern Instruments, UK). CMOS cameras, specifically C11440-50B, with an FL-280 Hamamatsu Photonics (Shizuoka, Japan) image capture sensor were used as detectors. Measurements were taken in a special cuvette for aqueous solutions; the cuvette was equipped with a 405 nm laser (version CD, S/N 2990491) and a sealing ring made of Kalrez material. Temperatures were taken with an OMEGA HH804 contact thermometer (Engineering, Inc., Stamford, CT, USA) for all measurements. Samples for analysis were detected and injected into the measuring cell using a 1 mL glass 2-piece syringe (tuberculin) using the Luer taper (Hamilton Company, Reno, NV, USA). To increase the statistical dose, the sample was pumped through the measuring chamber using a piezoelectric dispenser. Each sample was detected sequentially 6 times; the recording time was sequential and amounted to 60 s. For processing the footage of the Nanosight instrument, NTA 2.3 software applications (build 0033) and OriginPro program package were used; the Gauss function was used throughout as described previously [48,49]. The detailed steps of this work can be found in the Principle of Operation by B. Carr and A. Malloy [50,51]. The hydrodynamic diameter (D_h_) was calculated via the measurement of the two-dimensional Einstein–Stokes equation [52].

### 3.7. General Procedure for the CuAAC Reaction

In a 4 mL glass vial charged with phenylacetylene (70 mM), halogene derivatives (70 mM), sodium azide (84 mM), CuI (0.175–0.7 mM), and 0.001/0.1 mM macrocycles were deposited. After adding 1 mL of water, the resulting mixture was heated to 40 °C while stirring. The aqueous layers were extracted with hexane in the presence of NaCl. The triazoles were identified via GCMS using a calibration curve (halogen derivatives vs. concentration).

## 4. Conclusions

In the present work, a new series of macrocycles based on the classical calixarene was synthesized. The obtained calixarenes were successfully used in the catalysis of the CuAAC reaction. It has been shown that amphiphilic zwitterionic calixarenes exhibit aggregates with a predominantly cationic nature, and the stability of the aggregates formed by them depends on the length of the alkyl substituents of the lower rim. These macrocycles can be used as an efficient nanoreactor and substrate for copper iodide, which was evaluated in a single-reactor three-component system for the preparation of 1,4-disubstituted 1,2,3-triazole via a 1,3-dipolar cycloaddition of terminal acetylenes, halogen-containing derivatives of different structures, and sodium azide, where the use of an aqueous system, low temperature, and non-hazardous organic azide should be emphasized. Based on a comparative analysis of the calixarenes and TBAB/SDS, the higher selectivity of calixarene in the CuAAC reaction was established. This work opens a broad perspective for the possibility of utilizing calixarenes in the in CuAAC reaction.

## Data Availability

Data is contained within the article and Appendix A.

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
