# Peer review of "New Carboxytriazolyl Amphiphilic Derivatives of Calix[4]arenes: Aggregation and Use in CuAAC Catalysis"

_ijms, 2023, doi:10.3390/ijms242316663_

Round 1

Reviewer 1 Report

Comments and Suggestions for Authors

The manuscript is overall of suitable for the journal quality. I recommend minor revisions considering the following points:

1)                  Line 81-84: It is recommended to mention in the text in what solvent the spectra were measured. As seen from the caption to Figure 1 and from Figures S1-S4, the solvent was DMSO-d6. Note that in the experimental part DMSO/H2O water is mentioned as the solvent, but only for 1H spectra for some reason. For 13C it is stated to be pure DMSO. In that regard the following two comments arise:

2)                  -Are the carboxyl groups really subject to deuteroexchange with DMSO-d6 molecules?

3)                  -How does amphiphilicity of the molecules explain the broadening of 1H NMR spectra? Due to aggregation? Are there possible conformation equilibria that could also lead to briadening, and is there a way to distinguish between the two?

4)                  Line 91-93: In the interval of 160-167 ppm there are 5 signals in Figure 1 (b). For the other compounds, there is typically 3 signals in that range. Revision of the sentence is recommended

5)                  Line 135-139 and Table S1: It is recommended to provide explanation to the values presented in Table S1, particularly what are Dh, d1 and d2. The errors for d2 are often larger then the values themselves, which leads to question if the bimodal model was necessary here.

6)                  Line 135-139 and Table S1: the statement “when the critical aggregation concentration is reached, aggregates of 300 nm in size predominate” only appears somewhat valid for compound 7b, whereas other compounds possess completely different numbers in Table S1. Furthermore, there is no apparent monotonic correlation between the sizes and concentrations.

7)                  The NTA analysis section: as I have no expertise nor experience in this, I can’t validate this section, however, I would advise to include an analysis on agreement/disagreement between DLS and NTA.

8)                  Figure 2(a): Typo in the word aggregation

9)                  Line 169: organic substrates – perhaps a different word should be used for the calixarenes because substrates are typically understood to be the molecules which undergo catalytic transformation (PhCCH and BnCl in this case)

10)               Line 172-173: cases the concentration of CuI was 0.5 mole % - that is catalyst load, not concentration

11)               Table 3, reaction scheme: Substituent R is not necessary because it is always Bn group (the same in Table 4)

12)               Line 242-247: Synthesis – acetylenedicarboxylic acid is used in excess, and it is not clear how the unreacted material was removed from the product.

I attach pdf with markings on the mentioned phrases as well as several other possible typos that I was able to detect.

Comments on the Quality of English Language

Author Response

We thank Editor and Reviewers for comments and suggestions for improving the manuscript.

1)                  Line 81-84: It is recommended to mention in the text in what solvent the spectra were measured. As seen from the caption to Figure 1 and from Figures S1-S4, the solvent was DMSO-d6. Note that in the experimental part DMSO/H2O water is mentioned as the solvent, but only for 1H spectra for some reason. For 13C it is stated to be pure DMSO. In that regard the following two comments arise:

2)                  -Are the carboxyl groups really subject to deuteroexchange with DMSO-d6 molecules?

Response: The protons of the carboxyl group have increased acidity. As a result, deuterium-hydrogen exchange occurs in D2O or in a mixture of DMSO-d6 and D2O. In DMSO-d6, signals from the carbon atoms of carboxyl groups are also not observed due to the proton exchange that occurs both between the protons of neighboring carboxyl groups and with water molecules that are present in DMSO-d6 in trace amounts. The corresponding explanations are included in the text of the article on lines 81-83

3)                  -How does amphiphilicity of the molecules explain the broadening of 1H NMR spectra? Due to aggregation? Are there possible conformation equilibria that could also lead to briadening, and is there a way to distinguish between the two?

Response: The broadening of the signals in the 1H NMR spectra is caused by the fact that the resulting compounds are capable of aggregation in an aqueous medium. As for the conformation, it was previously shown (https://doi.org/10.3390/nano10061143) on simpler structural compounds that they are conformationally stable and exist in the cone configuration.

4)                  Line 91-93: In the interval of 160-167 ppm there are 5 signals in Figure 1 (b). For the other compounds, there is typically 3 signals in that range. Revision of the sentence is recommended

Response: The text of the article was corrected on lines 92-93. In the 13C NMR spectrum in the region of 167-160 ppm. 4 signals are observed: two from magnetically nonequivalent carbon atoms of the triazole ring and two from magnetically nonequivalent carbon atoms of carboxyl groups. Typically, signals from carbons 4 and 5 of the triazole ring have a chemical shift in the region of 124 ppm. and 143 ppm accordingly. However, in the case of compounds 7a-b and 8a-b, the triazole carbon atoms are associated with electron-withdrawing carboxyl groups, as a result of which the signals of these atoms are shifted to the low-field region.

5)                  Line 135-139 and Table S1: It is recommended to provide explanation to the values presented in Table S1, particularly what are Dh, d1 and d2. The errors for d2 are often larger then the values themselves, which leads to question if the bimodal model was necessary here.

Response: According to the mathematical model used in the Malvern software, the particle size variation is calculated from a correlogram of all particles, then related to a specific particle fraction as a percentage. This has the effect of making the uncertainty range larger than the obtained value. Thus, to adequately estimate the contribution of a bimodal system, the NTA method was used, which, due to the larger discretization of nanoparticles by size, allows us to avoid this limitation.

To avoid misunderstanding, we have removed the values of d1 and d2 from Table S1, leaving the values of PDI and Dh, the mean hydrodynamic diameter.

6)                  Line 135-139 and Table S1: the statement “when the critical aggregation concentration is reached, aggregates of 300 nm in size predominate” only appears somewhat valid for compound 7b, whereas other compounds possess completely different numbers in Table S1. Furthermore, there is no apparent monotonic correlation between the sizes and concentrations.

Response: This part was corrected.

7)                  The NTA analysis section: as I have no expertise nor experience in this, I can’t validate this section, however, I would advise to include an analysis on agreement/disagreement between DLS and NTA.

Response: NTA analysis is essentially a development of the DLS method. However, the main differences are the greater accuracy and size resolution of nanoparticles. In the case of polydisperse systems, for example, with a distributed structure of blocks with a diameter of 100 and 200 nm, the DLS method, due to the use of the correlation function as a basic method, will determine all particles in the solution as type 1 particles with a diameter of 150. That is why in the presented work it is used NTA method. For monodisperse samples, there is no difference between the sizes determined by the 2 methods, and in the case of polydispersity samples, the DLS method is not representative due to the high PDI associated with the statistics accumulation method. Therefore, this paper presents only NTA analysis, which ensures the reliability of the bimodal model. We revised the description of the results in the manuscript text obtained by the DLS method and justified the use of NTA method.

8)                  Figure 2(a): Typo in the word aggregation

Response: This typo was corrected.

9)                  Line 169: organic substrates – perhaps a different word should be used for the calixarenes because substrates are typically understood to be the molecules which undergo catalytic transformation (PhCCH and BnCl in this case)

Response: The phrase was corrected

10)               Line 172-173: cases the concentration of CuI was 0.5 mole % - that is catalyst load, not concentration

Response: The phrase was corrected

11)               Table 3, reaction scheme: Substituent R is not necessary because it is always Bn group (the same in Table 4)

Response: The scheme was corrected

12)               Line 242-247: Synthesis – acetylenedicarboxylic acid is used in excess, and it is not clear how the unreacted material was removed from the product.

Response: The description was added in the manuscript text:

After evaporation, unreacted acetylenedicarboxylic acid was remained in the reaction product. To remove it, the target compound was washed with 20 ml of diethyl ether. In this case, the reaction product was precipitated, which was then filtered and dried in vacuum.

Reviewer 2 Report

Comments and Suggestions for Authors

In this study, the authors designed and synthesized a novel class of calix[4]arenes that exhibit both high amphiphilic and zwitterionic properties. This was achieved by modifying the functional groups on the calix[4]arenes. The detailed characterization of the aggregation properties of these calix[4]arenes supports the authors' claims regarding their physical attributes. The use of these calix[4]arenes in catalyzing one-pot CuAAC reactions involving sodium azide, alkyne, and alkyl halide was also demonstrated. Notably, the authors highlight that the amphiphilic nature of the calix[4]arenes enhances reactivity in aqueous conditions, which are not inferior to the commercial catalyst TBABr and the surfactant SDS.

From my perspective, this work offers insightful structure-activity relationship studies on the aggregation and catalysis properties of these calix[4]arenes. It should attract broad interest from the readership of the International Journal of Molecular Sciences. Thus, I support the publication of this manuscript.

However, before publication, I recommend addressing a critical aspect. While the reactivity of calix[4]arenes is compared with TBABr and SDS, the manuscript lacks a comprehensive explanation of the working mechanism of calix[4]arenes in these reactions. A detailed discussion or comment on the mechanism is recommended.

Author Response

We thank Editor and Reviewers for comments and suggestions for improving the manuscript.

However, before publication, I recommend addressing a critical aspect. While the reactivity of calix[4]arenes is compared with TBABr and SDS, the manuscript lacks a comprehensive explanation of the working mechanism of calix[4]arenes in these reactions. A detailed discussion or comment on the mechanism is recommended.

Response: We have added the part of discussion related to the mechanism of this processes in the manuscript text.

The excellent catalytic abilities were attributed to the synergetic effect of charged fragments and cavities of calixarene skeleton  [https://doi.org/10.1007/s13738-011-0027-6, https://doi.org/10.1002/slct.201900902 ], where functionalized macrocycle unites properties of phase-transfer catalyst and micellar nanoreactor. The amphiphilic calixarenes are capable of both solubilizing and transferring hydrophobic substrates and the active center (copper iodide) from the aqueous medium into the cavity of the calixarene platform, as well as into the hydrophobic part of the formed associates.

However, this detail mechanism we will have studied in our further studies.

Reviewer 3 Report

Comments and Suggestions for Authors

The reviewed work depicts the synthesis of a fresh series of amphiphilic calix[4]arene derivatives, produced for the CuAAC reaction. The authors accurately determined the structures of the yielded organic compounds through suitable methods. The work comprises an extensive range of experiments and results. Subsequently, the suitability of the obtained compounds was tested as catalysts. In my view, the research is commendable, captivating and in line with the trend of "green chemistry" as it eliminates organic solvents, either partially or entirely. The use of water as a solvent is a significant step towards safeguarding the environment.

I have a few remarks, mostly on editorial inaccuracies.

·         The abstract should briefly describe the significant results obtained. Additionally, the authors provide information on the catalysis test conducted and compare it with classical catalysts. However, the result is missing.

·         On line 13, the first usage of "CuAAC" requires an explanation.

·         Line 40-41 requires explanations for the abbreviations THPTA, BTTAA, BTTES, TEOTA, BTTP, and BTTPS.

·         The positioning of the negative charge needs to be amended in structures 5a, 5b, 6a, 6b, 7a, 7b, 8a, and 8b in diagram 1.

·         Additionally, it is unclear why the authors display the C13 spectra for compounds 6b and 8a in Figure 1, rather than 6a and 8a or 6b and 8b.

·         Furthermore, line 153 to 154's statement "But dimers of the main fraction with hydrodynamic diameters of 206 (dimer 94) and 303 (dimer 140) are present in solution" should be reframed. In my opinion, dimer 94 or dimer 140 is not a sufficient explanation. There could be, for instance, a dimer of 94 nm.

·         In chapter 3, it is recommended to provide compound numbers as in the previous chapter and use bold for further descriptions (lines 217-226, 232-239).

·         Please correct cm-1 to cm-1 and C to °C.

·         In my opinion, it would be best to stick to the same units as before, which is why °C was used earlier, and also in line 316 and 327 (Kelvin).

The paper's introduction, purpose, and conclusion require revision to emphasise the crucial aspects and advantages (e.g. scientific, economic, or environmental) of using the mentioned compounds for catalysis.

In my opinion, the article is interesting, the literature used is up to date and can be published after taking into account the highlighted comments.

Author Response

We thank Editor and Reviewers for comments and suggestions for improving the manuscript.

I have a few remarks, mostly on editorial inaccuracies.

  • The abstract should briefly describe the significant results obtained. Additionally, the authors provide information on the catalysis test conducted and compare it with classical catalysts. However, the result is missing.

Response: The results of catalysis were added in abstract.

  • On line 13, the first usage of "CuAAC" requires an explanation.

Response: The explanation was added.

  • Line 40-41 requires explanations for the abbreviations THPTA, BTTAA, BTTES, TEOTA, BTTP, and BTTPS.

Response: The explanation of abbreviations was added.

  • The positioning of the negative charge needs to be amended in structures 5a, 5b, 6a, 6b, 7a, 7b, 8a, and 8b in diagram 1.

Response: The structures were corrected

  • Additionally, it is unclear why the authors display the C13 spectra for compounds 6b and 8a in Figure 1, rather than 6a and 8a or 6b and 8b.

Response: The Figure 1 was corrected.

  • Furthermore, line 153 to 154's statement "But dimers of the main fraction with hydrodynamic diameters of 206 (dimer 94) and 303 (dimer 140) are present in solution" should be reframed. In my opinion, dimer 94 or dimer 140 is not a sufficient explanation. There could be, for instance, a dimer of 94 nm.

Response: We thank the reviewer for asking such a insightful question. You are right: particles with a diameter of 303 nm can be either two stuck together particles with a diameter of 140 nm, or three with a diameter of 94 nm. It can also be an associate of nanoparticles with a diameter of 206 and 94 nm. In the case of classical DLS, it is almost impossible to answer this question without using a method such as cryo-TEM. However, taking into account the fact that the main fraction of nanoparticles, that is, the fraction that is most in the solution, is precisely nanoparticles with a diameter of 140 nm, we assume that it is they that stick together into particles with a diameter of 304 nm. However, it is not possible to exclude triple aggregation of particles with a diameter of 94 nm; therefore, changes have been made to the article.

In addition, references to the applicability of this method to similar systems are provided in the Materials and Methods section.

  • In chapter 3, it is recommended to provide compound numbers as in the previous chapter and use bold for further descriptions (lines 217-226, 232-239).

Response: lines 217-226 – In the sent version in doсx format, the substance numbers are in bold. However, when automatically converted to pdf format, the bold disappears. We'll upload our pdf file.

Response: Lines 232-239 – were corrected

  • Please correct cm-1 to cm-1and C to °C.

Response: The issue corrected

  • In my opinion, it would be best to stick to the same units as before, which is why °C was used earlier, and also in line 316 and 327 (Kelvin).

Response: The issue corrected

The paper's introduction, purpose, and conclusion require revision to emphasise the crucial aspects and advantages (e.g. scientific, economic, or environmental) of using the mentioned compounds for catalysis.

Response: The introduction, purpose, and conclusion of manuscript text were supplemented. The main goal of this work is to study the possibility of using calixarenes in the catalysis of the CuAAC reaction. Based on the compared analysis between calixarenes and TBAB/SDS, the higher selectivity of calixarenes in CuAAC reaction was established. This work opens a broad perspective for the possibility of utilizing calixarenes as additives in the CuAAC reaction. At this stage of our research, the benefits obtained relate only to the scientific field. However, we hope that our future research will achieve the economic and environmental benefits of calixarene in the CuAAC reaction.